# A New Crop Spectral Signatures Database Interactive Tool (CSSIT)

**Mohamad M. Awad [1,*], Bassem Alawar [2] and Rana Jbeily [3]**

[1] National Council for Scientific Research, Remote Sensing Center, Mansourieh 1107-2260, Lebanon
[2] Faculty of Agriculture, Avignon University, 84029 Avignon, France; bassem.al-awar@alumni.univ-avignon.fr
[3] Faculty of Science, American University of Science and Technology, Beirut 1100-2130, Lebanon; ranajbeily@hotmail.com
[*] Correspondence: mawad@cnrs.edu.lb

**Abstract:** In many countries, commodities provided by the agriculture sector play an important role in the economy. Securing food is one aspect of this role, which can be achieved when the decision makers are supported by tools. The need for cheap, fast, and accurate tools with high temporal resolution and global coverage has encouraged the decision makers to use remote sensing technologies. Field spectroradiometer with high spectral resolution can substantially improve crop mapping by reducing similarities between different crop types that have similar ecological conditions. This is done by recording fine details of the crop interaction with sunlight. These details can increase the same crop recognition even with the variation in the crop chemistry and structure. This paper presents a new spectral signatures database interactive tool (CSSIT) for the major crops in the Eastern Mediterranean Basin such as wheat and potato. The CSSIT's database combines different data such as spectral signatures for different periods of crop growth stages and many physical and chemical parameters for crops such as leaf area index (LAI) and chlorophyll-a content (CHC). In addition, the CSSIT includes functions for calculating indices from spectral signatures for a specific crop and user interactive dialog boxes for displaying spectral signatures of a specific crop at a specific period of time.

**Keywords:** spectroradiometer; database; query; crop mapping; spectral signature; hyperspectral; interactive

## 1. Summary

Since the invention of the portable field spectroradiometer in the late 1980s, a lot of research has been conducted on crop spectral behavior with visible and infrared electromagnetic spectrum [1]. Field spectroscopy in the laboratories is the measurement of very high spectral resolution radiance or irradiance in the field. It is useful to compute the reflectance (spectral signatures) of surface objects. The use of spectroradiometers in remote sensing data collection has many advantages over that of air and space borne sensors. Spectroradiometers can reduce the different known errors inherited in the data collected by space and air borne sensors, specifically atmospheric errors. The elimination of errors is due to the fact that the registration can remain fixed over the object for a much longer time with a short path length between the field instrument and the object being measured [2]. In addition, another advantage is the capability of providing higher spectral and temporal resolution with thousands of recorded data across the passive remote-sensing domains. The comparison of spectral signatures between different research papers is a problematic issue due to the many different techniques used for the capturing of spectral field data [3] and the influence of the sampling environment on the measurement (e.g., wind speed and direction, cloud cover and type, temperature, humidity, aerosols), viewing geometry (fore optic degree and the field of view or FOV and instantaneous-field-of-view or

IFOV, fore optic height above target, and ground), or illumination geometry (date, time, position and sun altitude, azimuth and orientation, smoke and haze). In addition, utilizing data from other studies requires an assessment of the data quality and suitability of the data set for the given task. The creation of a full database of crop spectral signatures is a complex task because it requires continuous monitoring and observations of the crops from seeding to harvesting. Currently, the existing spectral signature libraries are mostly made for non-crops analysis and research purposes. These libraries such as the Johns Hopkins University (JHU) Spectral Library [4], the Jet Propulsion Laboratory (JPL) Spectral Library [5], the United States Geological Survey (USGS-Reston) Spectral Library [6], the ASTER spectral library version 2.0 [7], and the ASU Spectral Library [8] include only spectral signatures of rocks, minerals, lunar soils, terrestrial soils, manmade materials, meteorites, some North American vegetation, snow, and ice.

There are other spectral signatures libraries designed for agriculture research purposes such as the SPECCHIO Spectral online database maintained by the Remote Sensing Laboratories in the Department of Geography at the University of Zurich [1,9]. This database displays metadata about instrument characteristics, date of acquisition, vegetation biophysical parameters, soil characteristics, and other important information. Moreover, the tool can display the spectral signature of major crops such as wheat and potato. The library has been tested, but there are many problems, such as no multi criteria query. A long list is displayed for each search which sometimes consists of hundreds of records. Instead, it should be more specific to the date and if possible to the location too. It should also specify more information about the crop itself for example plantation date of the crop. There are more libraries and databases for crop, vegetation and agriculture purpose, but some are no longer in service, such as the Vegetation Spectral Library, which was developed by the Systems Ecology Laboratory at the University of Texas in El Paso with support from the National Science Foundation [10].

The above reasons promoted the idea for developing a new crop spectral signatures database to improve crop management and to help establish precision in agricultural practices. In addition to the spectral signatures, the database includes a list of crop characteristics, by which a specific crop may be identified on an image. One of these characteristics is the Chlorophyll-a estimation which can help in the precise estimation of nitrogen fertilizer use as part of precision agriculture [11]. Another characteristic which can help in establishing precision in agriculture is to estimate plant water content using the library and remote sensing images. It may also provide valuable information to environmental and irrigation system managers to relieve dehydration symptoms and prevent permanent growth and production damage [12]. A new spectral signatures database is most useful if it is implemented as part of a software application that includes an attractive, effective, and friendly graphic user interface (GUI).

The tool can be deployed to map crops accurately. Normally, crop mapping requires tedious and continuous field work to verify and classify the satellite hyperspectral images. In the case of crop mapping, field work can become intolerable and expensive when crop classification requires images for different dates concurrently with crop growth stages. This is sometimes a necessary step in order to avoid differences in the date of plantation between different farmers. The problem can be solved if spectral signatures for crops at different growth phases are collected. Several important issues must be considered during the collection of data for building the spectral signature database such as the exact number of in situ measurements for obtaining the correct crop spectral signature and the sufficient number of times needed to collect data for each crop in different phenological stages. Another important and recommended matter for differentiating between different crops and for managing crops is to measure the physicochemical parameters for the crops during spectral signature acquisition. The most important physiochemical parameters are the leaf area index (LAI) and chlorophyll-a content (CHC). Jonckheere et al. [13] reviewed several direct and indirect methods for LAI estimation, and Gitelson [14] and Rossini et al. [15] reviewed several methods for CHC. Gitelson and Rossini concluded that these methods were very erroneous when applied to a specific crop, and their success depends on the type of crop, sowing date, and other issues such as climate. To overcome these problems,

physiochemical measurements must be done periodically and synchronously with spectral signature data collection. One of the possible uses of the parameters is to create a mathematical model for every major crop to predict these parameters from remote sensing data such as the one created for the chlorophyll-a content for wheat crop in [16].

In this research, an interactive tool (CSSIT) is developed. It is planned to include a spectral signatures database of major crops with physiochemical parameters. The tool will also include metadata for each survey (environmental conditions and others) and multi criteria search, retrieve, display, and analysis of the crop spectral signatures.

## 2. Data Description

To succeed in creating the new tool, several important issues must be considered. First we have to decide on the type of data to be collected such as spectral signatures, leaf area index, and chlorophyll-a content. The collection of these data is based on using an ASD FieldSpec 4 Hi-Res Spectroradiometer (Figure 1a) [17], SPAD 502 (Figure 1b) [18], and LI-3000C Portable Leaf Area Meter (Figure 1c) [19]. The 3 nm visible and near infrared (VNIR), 8 nm short wave infrared (SWIR) spectral resolution of the FieldSpec 4 Hi-Res spectroradiometer provides superior spectral performance across the full range of the solar irradiance spectrum (350–2500 nm). Combining spectral signatures, biophysical, and biochemical parameters can support modeling efforts by providing spatially and temporally distributed information on important vegetation characteristics, which would be very difficult to obtain otherwise. The LI-3000C sensor provides a powerful system for portable, non-destructive leaf area measurements. It utilizes an electronic method of rectangular approximation to provide 1 mm$^2$ resolution. Leaf area, leaf length, average width, and maximum width are logged by the readout console as the scanning head is drawn over a leaf. Leaf area index (LAI) is described as green leaf area per unit of ground area. It is a crucial biophysical parameter used in most mathematical models that often functions as the primary remote sensing based descriptor of vegetation density, phenology, and distribution across a landscape [20].

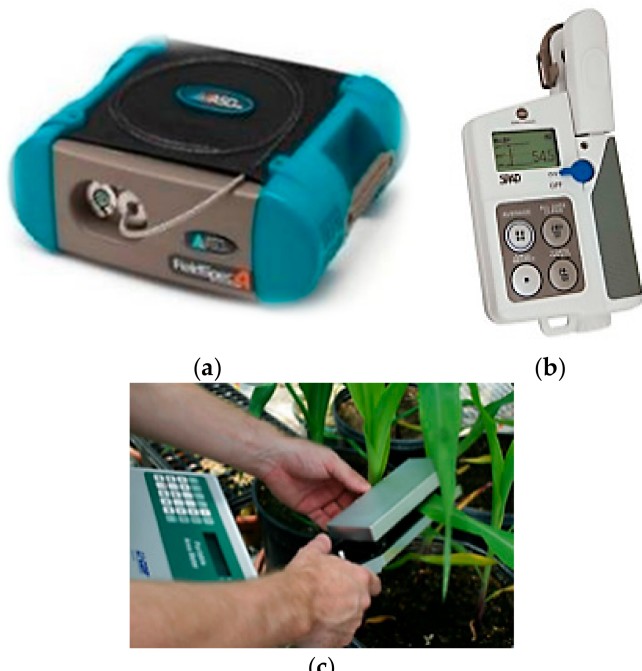

(**a**)　　　　　　　　　　　(**b**)

(**c**)

**Figure 1.** Devices for collecting crop related information. (**a**) Spectroradiometer, (**b**) SPAD 502, and (**c**) leaf area meter.

The SPAD 502 Chlorophyll Meter instantly measures the chlorophyll content or "greenness" of your plants to reduce the risk of yield-limiting deficiencies or costly over fertilizing. The SPAD 502 Plus quantifies subtle changes or trends in plant health long before they are visible to the human eye. The non-invasive measurement by the SPAD meter is simply done by clamping the meter over leafy tissue, and then by receiving an indexed chlorophyll content (CHC) reading (−9.9 to 199.9) in less than 2 s. Determining vegetation chlorophyll-a concentration is a recognized and regularly used method in agriculture applications for observing vegetation health. CHC is related to other vegetation parameters such as carotenoids, nitrogen, and maximum green fluorescence [21].

Sometimes it is necessary to combine the in situ biophysical parameter measurements with one time laboratory analysis. Direct chlorophyll-a measurement using chemical extraction is destructive, complex, and time-consuming. Sometimes the use of this method can help in finding a mathematical relationship between analyzed biophysical parameter values and the measured ones [16].

*Spectral Signatures and Parameters Collection Process*

An important research study [22] has recommended following planned procedures in field data collection, especially in collecting crop spectral signatures. The procedures should include specific date and time, percentage of cloud cover, temperature, humidity, homogeneity of species, health of species, size of the surveyed area, illumination and viewing setting, device calibration, and instrument settings. In this work, several agricultural areas (field crops) were selected for data collection based on the crop type, crop pattern, area size, and accessibility. Many samples of spectral signatures are collected from selected plots of size $1 \times 1$ m$^2$ in each crop field. The number of plots depends on the size of the crop field. In general, three to five plots are selected randomly for each $100 \times 100$ m$^2$. The intensive field trips have allowed us to collect the spectral signatures of major crops such as potato, wheat, onion, vineyards, fava bean, fruit trees, and alfalfa. Before registering spectral signatures, the spectroradiometer is calibrated using dark current and white reference measurements using a white reference panel (Spectralon). The resolution of the spectroradiometer in the visible to short wave infrared ranges between 3 (nm) to 8 (nm). Each registered spectrum consists of 4 individual measurements recorded consecutively at 15 s intervals and averaged by the ASD instrument. In addition, the biophysical parameters collection process is done as follow:

*i*- Concerning LAI, a sample of each plant in the plot is extracted from the root after measuring the area covered by the plant. LAI is explained in the following Equation:

$$LAI_{plant} = \frac{\sum_{k=1}^{Nl} LAv_k}{C_{area}}, \tag{1}$$

where $LAv_k$ is the area of leaf $k$ for a specific plant measured using the LI-3000C Portable Leaf Area Meter, $Nl$ is the number of leaves per plant, and $C_{area}$ is the area occupied by the plant on the soil.

*ii*- For each plant the CHC for selected leaves are measured using the SPAD 502 device. The final CHC in SPAD unit is obtained by averaging the measured leaves CHC values (Equation (2)). Complete details about the conversion from SPAD units to total chlorophyll per unit leaf area (nmol/cm$^2$) can be found in [14]. This includes type of used calibration curves.

$$CHC_{plant} = \frac{\sum_{i=1}^{n} CHC_i}{n} \tag{2}$$

The average CHC is computed based on measuring SPAD units of $n$ leaves per plant.

Based on intensive field work, taking 3 to 5 samples per field is recommended where each sample represents a plot in a field crop with an area size equal to at least one hectare. The surveys should account for different natural and environmental conditions such as soil type, soil temperature, vegetation temperature, climate conditions, and topography. Regarding the time of data acquisition, the field spectroscopy protocols [23] recommend that the measurement time must be as close as possible

to solar noon. In our case, most of the field work is done between 10:30 in the morning and 2:30 in the afternoon. It is important that in order to create unique and measureable spectral signatures for crops, spectral signatures databases must consider the spatiotemporal changeability of the vegetation. Field surveys should be done on different phenological cycles for each crop and to detect differences in LAI, CHC, and other physiochemical parameters that can help distinguish between different crops and to create unique signature for specific crop. In our case and depending on the climate conditions, the trips are conducted once every week such that each week a different area is visited and surveyed. This pattern of visits helps in covering a large area of study in one month or less. The same procedure is repeated and completed before harvesting of the crops. The total number of collected spectral signatures after quality check was large enough (>100), which helped in understanding the phenology of each studied crop. Sometimes, spectral signatures are collected for non-crop bodies such as soil, water, and others.

## 3. Methods

The creation of the tool requires analysis and design of user requirements. In other words, organized and planned steps are conducted to create the database management system (DBMS) and link it to thematic functions of the tool such as displaying spectral signature graph, calculation of vegetation indices, resampling of spectral signatures, and showing the biophysical parameters. This includes a user friendly interface with easy to use dialog boxes in the final tool. Figure 2 shows a flowchart that explains the different steps required to implement the crop spectral signatures database interactive tool (CSSIT).

Information is displayed as graphs of spectral signatures, tables listing different wavelengths and reflectance values, crop photos at the time of data acquisition, and much more information.

To populate the database with data and metadata, the task of collecting field samples starts before the tool implementation, and it is based on user requirements (farmers, decision makers, and researchers) and must continue to do so in the future.

Normally, after the requirement analysis and collection phase, the project leader and the development team can start to implement the conceptual design where one of the expected results is the entity relationships (ERs). Figure 3 shows the relationship between different entities that make up the database management system (DBMS). As an example, in the "metadata" table the item "NameofFile" refers to the name of the file which contains the spectral signature of the crop.

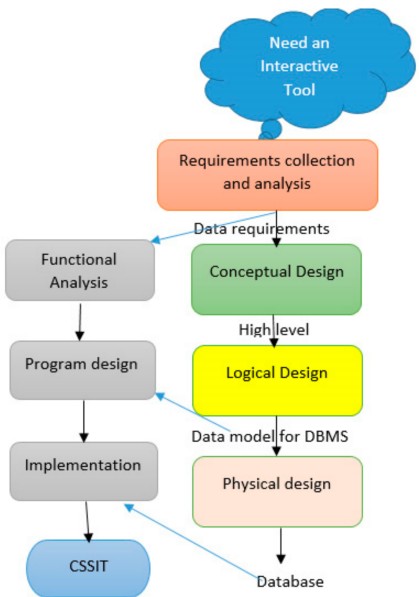

**Figure 2.** The different steps to build the new interactive tool.

Several metadata can be recorded such as the spectroradiometer's field of view (FOV), sky status, surveillance angles, time of data collection, and supplementary crop parameters. In addition, important issues must be recorded such as file name and file size on the disk, format for recording radiance or reflectance, and whether any derivative is applied.

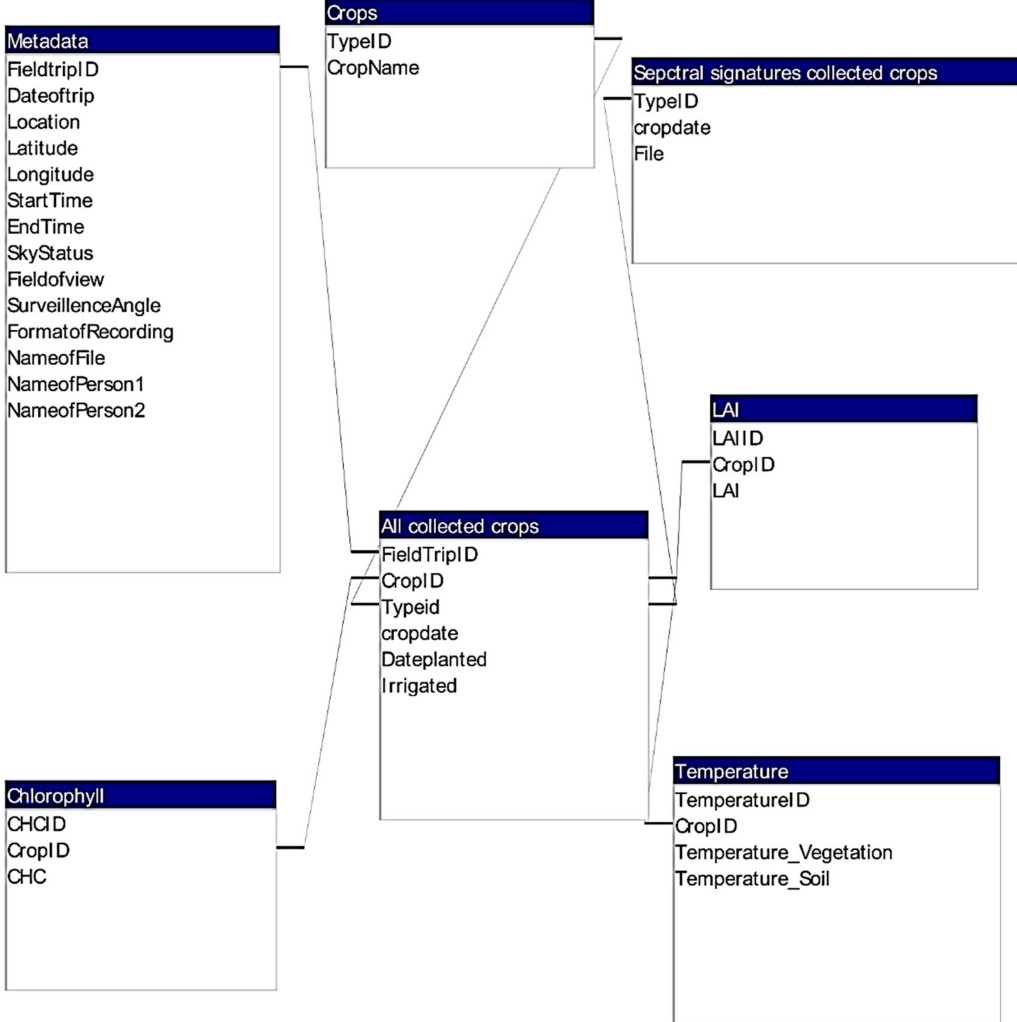

**Figure 3.** Simple entity relationship (ER) for the initial CSSIT's database.

Significant progress has been achieved and an initial interface has been created to be deployed on the Web in the future.

The tool allows guest users for limited access. Special users are registered users who can have wider access to specific areas and features of the database. Finally, there is an administrator account that can modify content of the database and can add more collected data. Users with more privileges may have the right to backup and restore the database and maintain the tool. Figure 4a–d show some interactive dialog boxes in CSSIT.

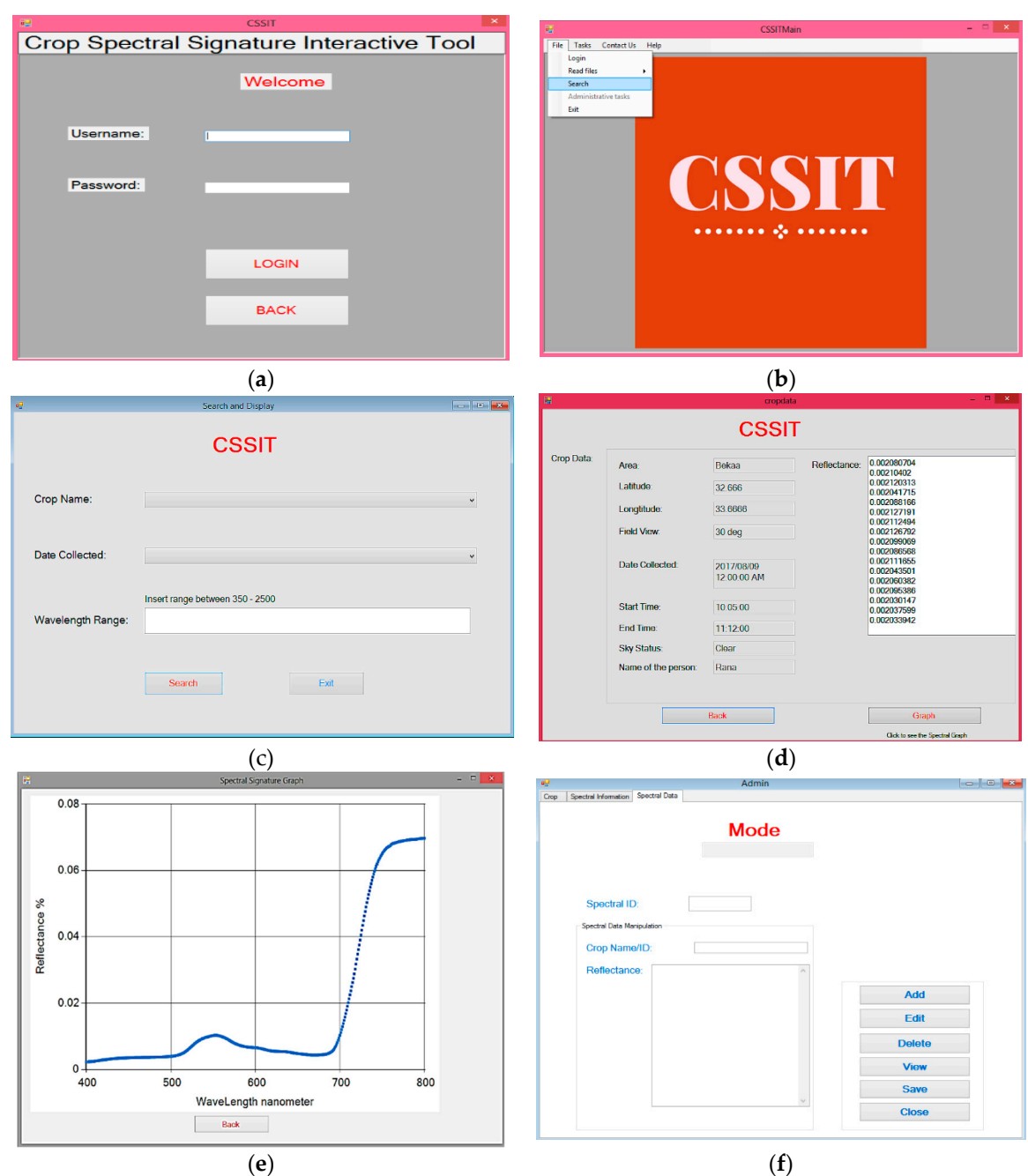

**Figure 4.** General view of the CSSIT windows. (**a**) Login dialog box, (**b**) main interface, (**c**) search dialog box, (**d**) metadata dialog box, (**e**) displayed spectral signature, and (**f**) administrator tasks.

As can be seen from the above figures when the application is launched, it requests authorization in order to allow the user to access its database and other existing functions and features. After gaining access to the application (CSSIT), the guest user can select search item from the main interface window to launch the search dialog box. After searching for a specific crop and date of collection, one can either display the metadata of the crop or/and display the graph of the crop's spectral signature. If the user is authorized as an administrator then the tasks dialog box can be displayed to add, edit, or remove records. Finally, although the application does not allow complete access for guest or normal authorized users, in the future when it is deployed on the Web more features will be added such as a feedback and suggestions dialog box. A copy of the main application CSSIT under the name

"setup.zip" is made available with this research paper in Supplementary Materials. The database is not included, but an update of the application may be provided to the journal in the near future.

*3.1. Vegetation Indices and CSSIT*

The vegetation indices (VIs) are combinations of surface reflectance at two or more wavelengths designed to highlight a particular property of vegetation. Each of the VIs is designed to emphasize a particular vegetation property [24].

More than 150 VIs have been published in scientific literature, but only a small subset have substantial biophysical basis or have been systematically tested [25]. The VIs are normally divided into several categories depending on their use. However, the most known categories are the broadband and narrowband VIs [26]. Because we are using very high resolution spectral data, it has been decided that some narrow band VIs can be used in CSSIT. Examples of what VIs can be calculated using CSSIT (Figure 5) are the red edge normalized difference vegetation index (RENDVI) (Equation (3)) [27] and the narrow bands (hyperspectral) normalized difference nitrogen index (NDNI) [28]. This index is designed to estimate the relative amounts of nitrogen contained in vegetation canopies (Equation (4)).

$$RENDVI = \frac{\rho_{750} - \rho_{705}}{\rho_{750} + \rho_{705}}, \tag{3}$$

$$NDNI = \frac{\log\left(\frac{1}{\rho_{1510}}\right) - \log\left(\frac{1}{\rho_{1680}}\right)}{\log\left(\frac{1}{\rho_{1510}}\right) + \log\left(\frac{1}{\rho_{1680}}\right)}, \tag{4}$$

where $\rho_{705}$, $\rho_{1510}$, $\rho_{1510}$, and $\rho_{1680}$ are the reflectance values for wavelengths 705, 750, 1510, and 1680 nm respectively. Further important indices that can be computed using CSSIT are the transformed chlorophyll absorption reflectance index (TCARI) [29], modified triangular vegetation index (MTVI) [29], modified triangular vegetation index—improved (MTVI2) [29], Vogelmann red edge index 1 (VREI1) [29], and Vogelmann red edge index 2 (VREI2) [29]. The following equations are used to calculate these indices.

$$TCARI = 3\left[[\rho_{700} - \rho_{670}] - 0.2[\rho_{700} - \rho_{550}]\left[\frac{\rho_{700}}{\rho_{670}}\right]\right], \tag{5}$$

$$MTVI = 1.2[1.2[\rho_{800} - \rho_{550}] - 2.5[\rho_{670} - \rho_{550}]], \tag{6}$$

$$MTVI2 = \frac{1.2[1.2[\rho_{800} - \rho_{550}] - 2.5[\rho_{670} - \rho_{550}]]}{\sqrt{(2\rho_{800} + 1)^2 - \left(6\rho_{800} - 5\sqrt{\rho_{670}}\right) - 0.5}}, \tag{7}$$

$$VREI1 = \frac{\rho_{740}}{\rho_{720}}, \tag{8}$$

$$VREI2 = \frac{\rho_{734} - \rho_{747}}{\rho_{715} + \rho_{726}}, \tag{9}$$

where $\rho_{550}$, $\rho_{670}$, $\rho_{700}$, $\rho_{715}$, $\rho_{720}$, $\rho_{726}$, $\rho_{734}$, $\rho_{740}$, $\rho_{747}$, and $\rho_{800}$ are the reflectance values for wavelengths 550, 670, 700, 715, 720, 726, 734, 740, 747, and 800 nm respectively. These are just a few applications of the CSSIT to extract useful information about crops by calculating VIs. The calculation of a specific VI is done by the user and is based on the selection of specific crop and the date of spectral signature collection. In the next section, several experiments are conducted to illustrate and emphasize the importance of the CSSIT in crop monitoring.

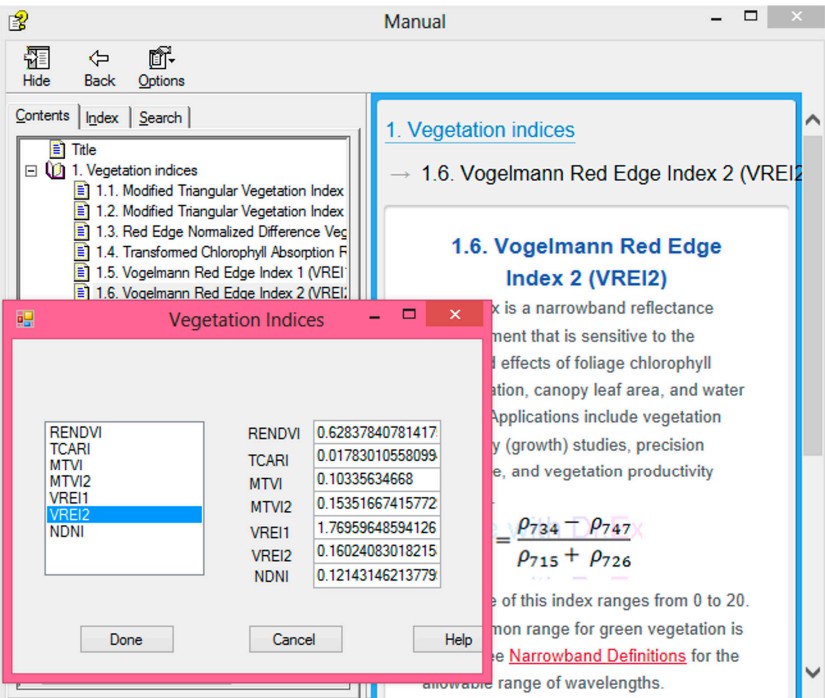

**Figure 5.** Indices calculation dialog box with help window.

### 3.2. Resampling of Spectral Signatures and Other Analysis

The database can be used by experts to study crop characteristics and crop behavior during growth stage. Several spectral signatures for the same crop at different periods of time can be retrieved from the database using the tool to study the interaction of the crop with the incident electromagnetic waves (Figure 6). One can notice in the graph as the time progresses that as the wheat's leaves grow the reflectance percentage of near infrared increases. This is due to the increase in the area size of the leaves, and it means that the photosynthesis of the crop is higher. At the end, when the wheat crop approach harvesting time, the reflectance diminishes and the leaves turn a yellow color.

A lot of important information can be obtained using the tool. One can use different biophysical parameters and the surrounding environment such as soil temperature and crop temperature to check the strength of correlation between chlorophyll-a content and the temperature.

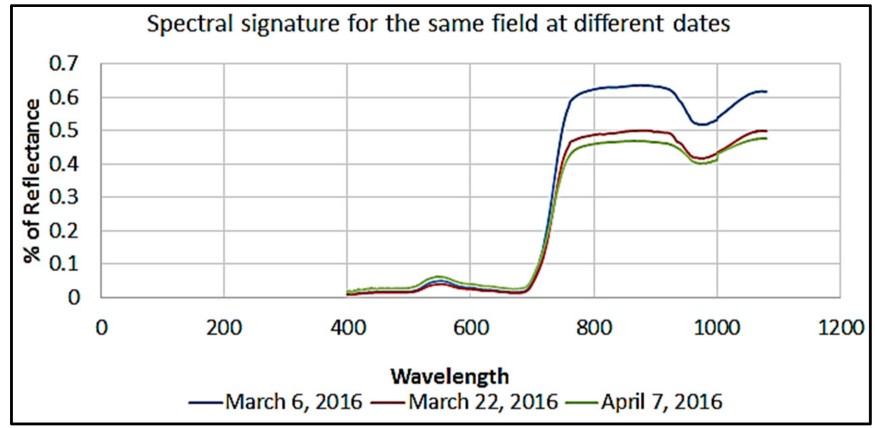

**Figure 6.** Spectral signatures of winter wheat.

The graphs in Figure 7a,b show more than 20 survey points. These points are used in the regression analysis between soil temperature and CHC. The correlation between temperature and CHC is stronger with respect to the crop than with respect to soil temperature. It is possible also to conduct research on the relationship between different wavelengths of the reflected spectrum from the visible to short wave infrared and the biophysical parameters. It is possible to check the strength of the correlation between chlorophyll-a content measured using a SPAD meter and some VIs, which are computed using specific electromagnetic wavelengths. Figure 7c,d show graphs of the relationship between calculated TCARI, NDNI, and measured SPAD units.

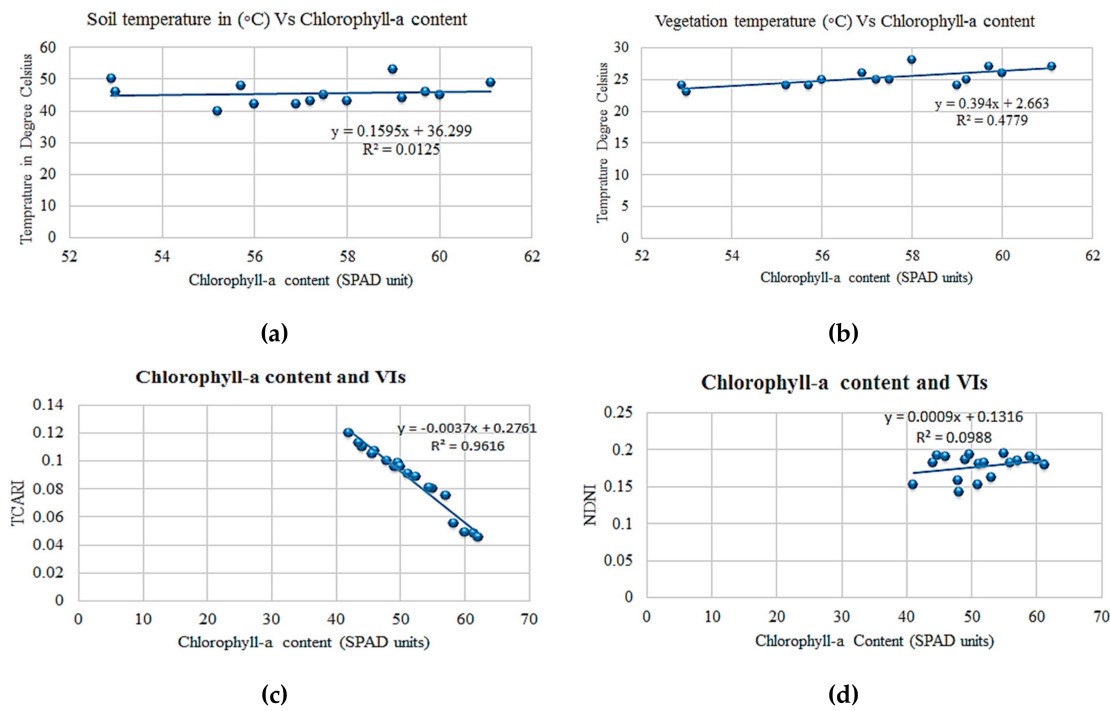

**Figure 7.** Chlorophyll-a content in relationship to (**a**) soil temperature, (**b**) vegetation temperature, (**c**) TCARI, and (**d**) NDNI.

Normally, the physiochemical data are collected synchronously with spectral signature collection, and they are entered in the database as shown in Table 1. The spectral signatures in the database can be resampled to fit the spectral resolution of any remote sensing hyperspectral aerial or satellite image. Although this step is not yet automated in the tool, a huge effort is being made to include this functionality in the tool. It is expected that the tool will be able to resample all existing hyperspectral satellite data. Figure 8a–c show examples of these resampling results. In the figures one can see the original image resampled to the Compact High Resolution Imaging Spectrometer (CHRIS) on the Proba satellite platform [30–32]. CHRIS-Proba mode 1 has 63 bands with spectral resolution that varies between 2 to 20 nanometers and covers only the wavelength from 406 to 1003 nanometers. Using Proba steering capabilities in along and across track directions enables observation of selectable targets well outside the nominal field of view of 1.3°. Images will generally be acquired in sets of 5, these being taken at along track angles of ± 55 degrees, ± 36 degrees, and as close to nadir as possible.

The spectral signatures are resampled again to fit the spectral resolution and wavelengths which are covered by a Hyperion hyperspectral imager on the EO-1 satellite platform [33,34]. A Hyperion image consists of 242 bands, has spectral resolution of 10 nanometers, and covers the entire reflected electromagnetic wavelength (350 to 2500 nanometers). The stability of the overpass time of EO-1 from 2001 to 2008 helped Hyperion to capture at and off nadir images [35].

**Table 1.** Samples of collected physiochemical data collection.

| Crop | Longitude | Latitude | SPAD Meter | Leaf T$^0$ | Soil T$^0$ |
|---|---|---|---|---|---|
| Potato | 35 48.798 | 33 46.421 | 49.4 | 22 | 30–33 |
|  |  |  | 41 | 21 | 30–33 |
|  |  |  | 44.2 | 20 | 30–33 |
|  |  |  | 45.8 | 25 | 30–33 |
| Potato | 35 48.799 | 33 46.396 | 46.5 | 23 | 36 |
|  |  |  | 37.9 | 22 | 36 |
|  |  |  | 43.6 | 20 | 36 |
|  |  |  | 42.9 | 21 | 36 |
| Potato | 35 48.367 | 33 45.428 | 49.3 | 20 | 28 |
|  |  |  | 41.7 | 18 | 28 |
|  |  |  | 46.7 | 20 | 28 |
|  |  |  | 44.8 | 20 | 32 |
| Potato | 35 48.894 | 33 46.687 | 36.8 | 21 | 20–22 |

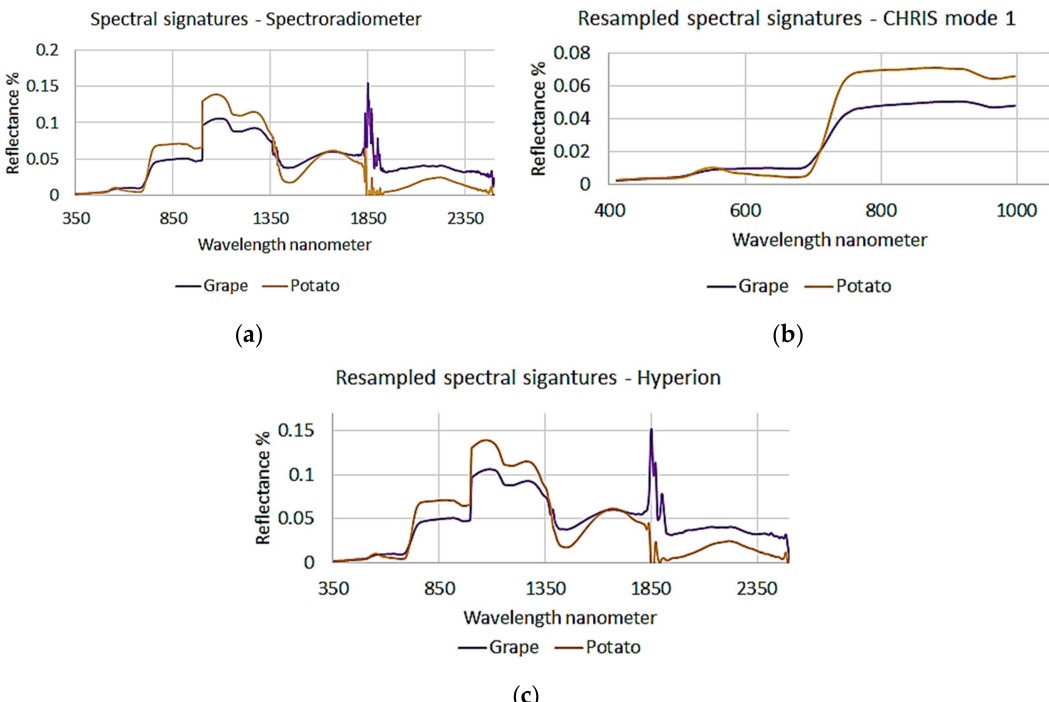

**Figure 8.** Spectral signatures. (**a**) Spectroradiometer, (**b**) resampled to CHRIS-Proba, and (**c**) resampled to Hyperion.

## 4. User Notes

In the monitoring and mapping process of natural resources and agricultural products, a reliable geospatial technology supported by an up to date information database and metadata is needed. The advancement in technology represented by the existence of a reliable tool that can increase the efficiency of geospatial information extraction and reduce errors in decision making, can have a positive return for farmers. In this research, a new crop spectral signatures database interactive tool (CSSIT) is developed and in the enhancement phase. The objective of the tool is to help in the improvement process of crop mapping and in turn to ease the crop yield estimation task. It is shown in this paper that significant steps have been achieved toward completing the interactive database tool. In addition, the tool includes more ancillary information about the physiochemical parameters for some major crops in the future. The tool can include mathematical models for estimating some parameters such as

chlorophyll-a content for some crops such as wheat. The new tool can help in better understanding the crop growth process and this is proved in some experimental results such as checking the strength of the relationship between temperature of soil, vegetation, and chlorophyll-a content.

It is expected that the work will continue in the future by developing more features for the tool based on the needs of the farmers and the experience that will be obtained as a result of periodical contact with farmers and experts. It is also anticipated to include more details about crop spectral signatures of existing and new species. Finally, it is also planned to include the functions previously discussed in the methods such as a resampling process for different hyperspectral satellites and aerial sensors.

**Supplementary Materials:** The following are available online at http://www.mdpi.com/2306-5729/4/2/77/s1, Software: CSSIT–Main application.

**Author Contributions:** M.M.A. is the project coordinator and is responsible for the software and database analysis, design, and programing. B.A. has the mission of data entry, data collection, and testing the tool. Finally, R.J. is a computer science graduate whose main tasks are programing and creating the database.

**Funding:** The author would like to thank the Lebanese CNRS for supporting this ongoing research "Creating an interactive crop spectral signatures database for Lebanon" under the Research Grant Program.

**Conflicts of Interest:** The authors declare no conflicts of interest.

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
