# Peer review of "A New Crop Spectral Signatures Database Interactive Tool (CSSIT)"

_data, 2019_

Reviewer 1 Report

The manuscript entitled "A NEW CROP SPECTRAL SIGNATURES DATABASE INTERACTIVE TOOL (CSSIT)" presents an interesting approach for spectral profiles analysis. The paper goes for a straightforward way in which explains some of the parameters that can be adjusted relating also the novelty from other data sources. Since this is my first review for Data Journal, it took me a bit more time to review this paper and to understand the goal of the Journal. However, by searching for the title, I found the DOI: 100.13140/RG.2.2.35595.80164 that presents exactly the same figures shown in this manuscript. I wonder if this allowed since there is no mention in the paper. In this sense, a good explanation must be given.

In general, the manuscript is too superficial and can be enhanced. The word "agriculture sector (L8)" could be changed to commodities in which the agriculture sector plays an important role. Another issue is the food security that could be cited too. The paper itself mentioned several physical and chemical parameters from crops (L17). However, it would be worthy to know how many species are available and how many growth stages are considered. The number of growth stages, mainly those that are critical from certain cultures are fully neglected. If not sampled in the field, the word "expected" could be added in L19. Interestingly for the proposed dataset are the several forthcoming hyperspectral missions that have several spectral bands and therefore very close to field measurements. Therefore, the paper could be substantially improved.

Some other comments:

L26: change "crop reaction with..." to spectral behavior in the visible... (the paper should be carefully revised for such terms/ I will point out only once);

L27: add laboratory (because you can be used it for validation purposes too);

L31: add "and adjacent contribution";

L56: please rephrase;

L66: please explain;

L74: which dataset? please explain;

L75: which images? please explain;

L76: in which situation?

L89-91: please improve writing;

L93: it is still not clear that it will be available? 

L94: what about sun and viewing geometry? among other related with planting orientation, density, environment conditions, genotype among others?

Fig1: improve the quality of the figure and also size;

L140: use nmol*cm^-2;

Fig4: CSSIT windows on a PC? online? how it works effectively? how can a single user contribute to the system? is it allowed?

L194: please start with band ratios and also NDVI (it was the first index);

Fig4: NDVI is even not listed in the CSSIT interface, why?

L227: please explain, authors just present results and don't provide any explanation;

Fig.5: please follow a standard format (April 14, 2016);

Fig.6: missing caption and also discussions;

Final remarks section doesn't exist; it could be added listing all minor contributions and then the statement for the contribution and novelty aspect of the paper;

Author Response

Dear Sir,

Thank you for your support and efforts. Your comments have
improve my paper significantly. I followed all your remarks to correct all mistakes. Please note that all corrected mistakes and added remarks are written in red. Some redundant information and/or references are also removed.

Best regards

The manuscript entitled "A NEW CROP SPECTRAL SIGNATURES DATABASE INTERACTIVE TOOL (CSSIT)" presents an interesting approach for spectral profiles analysis. The paper goes for a straightforward way in which explains some of the parameters that can be adjusted relating also the novelty from other data sources. Since this is my first review for Data Journal, it took me a bit more time to review this paper and to understand the goal of the Journal. However, by searching for the title, I found the DOI: 100.13140/RG.2.2.35595.80164 that presents exactly the same figures shown in this manuscript. I wonder if this allowed since there is no mention in the paper. In this sense, a good explanation must be given.

A-    Thank you for your remark. The title is found on Researchgate which is the name of the project which includes
a progress reports and preprint. The paper has not been published anywhere. More details can be found in the following Web page. In addition, an abstract was submitted to local non referred conference.

https://www.researchgate.net/publication/325387217_CSSIT_a_new_crop_spectral_signatures_interactive_tool

In general, the manuscript is too superficial and can be enhanced. The word "agriculture sector (L8)" could be changed to commodities in which the agriculture sector plays an important role.

A-    The line is changed and it is highlighted in red thank you.

 Another issue is the food security that could be cited too.

A-    Another line (in red) is added in the abstract concerning food security it is highlighted in red thank you.

The paper itself mentioned several physical and chemical parameters from crops (L17). However, it would be worthy to know how many species are available and how many growth stages are considered.

The number of growth stages, mainly those that are critical from certain cultures are fully neglected.

 If not sampled in the field, the word "expected" could be added in L19.

A-    The paragraph from Line 17 to line 22 is rewritten in red. The new paragraph
reflect the ideas the reviewer proposed thank you.

Interestingly for the proposed dataset are the several forthcoming hyperspectral missions that have several spectral bands and therefore very close to field measurements. Therefore, the paper could be substantially improved.

A-    The idea is inserted in the user notes in red thank you

Some other comments:

L26: change "crop reaction with..." to spectral behavior in the visible... (the paper should be carefully revised for such terms/ I will point out only once);

A-    Corrected and highlighted in red thank you

L27: add laboratory (because you can be used it for validation purposes too);

A-    Corrected and highlighted in red  thank you

L31: add "and adjacent contribution";

A-    Added and highlighted in red  thank you

L56: please rephrase;

A-    Rephrased and highlighted in red thank you

L66: please explain;

A-Rephrased to be clearer and more explainable and highlighted in redthank you

L74: which dataset? please explain;

A-The sentence is rewritten, clarified, and highlighted in red        thank you.

L75: which images? please explain;

A-The sentence is corrected and highlighted in red

L76: in which situation?

A-    More information is added and highlighted in red

L89-91: please improve writing;

A-    Writing of the sentences has been improved and highlighted in red

Gitelson and Rossini concluded that these methods were very erroneous when applied to a specific crop, and their success depends on the type of crop, date planted and other issues such as climate. To overcome these problems, physiochemical measurements must be done periodically in synchronous with spectral signatures data collection.  One of the possible uses of the parameters is to create a mathematical model for every major crop to predict these parameters from remote sensing data such as the one created for the chlorophyll-a content for wheat crop in [16].

L93: it is still not clear that it will be available? 

A-    The sentence is rewritten and highlighted in red

In this research an interactive tool (CSSIT) will be developed. It is planned to include spectral signatures database of major crops with physiochemical parameters. The tool will also include metadata for each survey and multi criteria search, retrieve, display and analysis of the crop spectral signatures.

L94: what about sun and viewing geometry? among other related with planting orientation, density, environment conditions, genotype among others?

A-    Corrected thank you

We added the following sentence The tool will also include metadata for each survey (environmental conditions and others)

Fig1: improve the quality of the figure and also size;

A-    Size and quality are improved thank you.     

L140: use nmol*cm^-2;

A-    Changed thank you.

Fig4: CSSIT windows on a PC? online? how it works effectively? how can a single user contribute to the system? is it allowed?

A-    Thank you for your precious remarks. More dialog boxes are added with higher resolution and the following is added

As can be seen from the above figures when the application is launched, it requests authorization in order to allow the user to access its database and other existing functions and features. After gaining access to the application (CSSIT), the guest user can select search item from the main interface window to launch the search dialog box. After searching for a specific crop and date of collection, one can either display the metadata of the crop or/ and display the graph of crop’s spectral signature. If the user is authorized as an administrator then tasks dialog box can be displayed to add, edit, or remove records. Finally, although the application does not allow complete access for guest or normal authorized users, but in the future when it is deployed on the Web more features will be added such as feedback and suggestion dialog box. 

L194: please start with band ratios and also NDVI (it was the first index);

Fig4: NDVI is even not listed in the CSSIT interface, why?

A-    Thank you for your important remarks. IN this application only some narrowband VIs are use. The following is added in the text:

The VIs are normally divided into several categories depending on their use. However, the most known categories are the broadband and narrowband VIs [26]. Because we are using very high resolution spectral data it is decided that some narrow band VIs can be used in CSSIT.  As an example of what VIs can be calculated by CSSIT (Figure 4) is Red Edge Normalized Difference Vegetation Index (RENDVI) (eq. 3.) [27], the narrow bands (hyperspectral) Normalized Difference Nitrogen Index (NDNI) [28]. This index is designed to estimate the relative amounts of nitrogen contained in vegetation canopies (eq. 4).

                                                                                                                                                                               (3)

[26] D. Sims, and J. Gamon., Relationships between Leaf Pigment Content and Spectral Reflectance Across a Wide Range of Species, Leaf Structures and Developmental Stages, Remote Sensing of Environment 2002, vol. 81, pp.337-354.

[27] A. Agapiou, D. Hadjimitsis, D. Alexakis, Evaluation of Broadband and Narrowband Vegetation Indices for the Identification of Archaeological Crop Marks, MDPI Remote Sensing journal 2012, vol. 4, pp. 3892-3919.

L227: please explain, authors just present results and don't provide any explanation;

A-    Explanation are added as follow:

One can notice in the graph as the time progress that as the wheat’s leaves grow the reflectance percentage of near infrared increase. This due to the increase in the area size of the leaves and which means that the photosynthesis of the crop is higher. At the end when the wheat crop approach harvesting time the reflectance diminishes and the leaves turns into yellow color.

Fig.5: please follow a standard format (April 14, 2016);

A-    Graph changed according the suggested format thank you

Fig.6: missing caption and also discussions;

A-    Caption is added thank you.

Final remarks section doesn't exist; it could be added listing all minor contributions and then the statement for the contribution and novelty aspect of the paper;

A-    Thank you for your remarks the following is added in a separate section

In the monitoring and mapping process of natural resources and agricultural products, a reliable geospatial technology supported by an up to date information database and metadata is needed. The advancement in technology represented by the existence of a reliable tool which can increase the efficiency of geospatial information extraction and which can reduce errors in decision making, can have positive return to farmers. In this research a new crop spectral signatures database interactive tool (CSSIT) is developed and in the enhancement phase. The objective of the tool is to help in the improvement process of crop mapping and in turn to ease the crop yield estimation task. It is shown in this paper that significant steps have been achieved toward completing the interactive database tool. In addition, the tool includes more ancillary information about the physiochemical parameters for some major crops. The tool can include in the future mathematical models for estimating some parameters such as chlorophyll-a content for some crops such as wheat. The new tool can help in better understanding the crop growth process and this is proved in some experimental results such as checking the strength of the relationship between temperature of soil, vegetation and chlorophyll-a content.

It is expected that the work will continue in the future based on the need of the farmers and the experience which will be obtained as a result of periodical contact with farmers and experts. It is also anticipated to include more details about crops spectral signatures of existing and new species. Finally it is also planned to include the functions discussed before in the methods such as resampling process for different hyperspectral satellites and aerial sensors.

Reviewer 2 Report

I have reviewed the manuscript data-505923 and I believe the authors have developed an interesting interactive tool (CSSIT) which includes spectral and physiochemical parameters, and  allow to perform different tasks with the data (search, visualization, analysis…).

I consider that there are some details that should be improved in order to a better understanding of the manuscript. I have some suggestions to improve it. My comments are detailed below:

Mayor comments:

1.      L98-99. This sentence is very simple. I suggest including more information about these devices (only with the name is very difficult to know the type of measure (i.e. SPAD 502)). Additionally, in order to a better reading, I suggest to move the comments of LAI (L102-105) and CHC (L 105-108) in the moment the authors comment each device.

2.      Authors should explain more specifically the data collection process (Section 2.1). As authors comment, the data collection should be planned controlling different variables (environmental, agronomic and technical variables). Nevertheless, the subsequent explanation (L124-129) comments superficially only some characteristics of the data collection characteristics. Authors use twice the word “Several” and there is no specific information about the number of crops and their characteristics, and the environmental conditions of the data collection. Additionally, the comment relate to the time of data acquisition (L147-148), should be located when the procedure of data collection is mentioned.

3.      In section 3.1, the authors comment that some vegetation indices can be calculated with CSSIT (but there is no specific information about that in the manuscript). Only observing the figure of page 7 I can know the vegetation indices (6) programmed. Nevertheless, in the manuscript there are two examples of the indices. I consider authors should include the six vegetation indices (not only two), with the equation and the reference. This information can be shown in a table.   

4.      L 229-231. Authors comment that correlation studies can be performed with CSSIT. IS the information about soil temperature or crop temperature included in the database. I can not see this information in Figure 3. If there are more variables that are in the database but are not shown in Figure 3, I suggest including a table with all variables available in the database.

5.      Figure of regression analysis. Are these graphs directly obtained with CSSIT? Or can any registered user download the data to perform any analysis with other software? There is no information about that in the manuscript.

Minor comments:

1.      The style of this journal does not suggest the title in capital letters.

2.      L 23. Remove the full point.

3.      L25-26. Revise the sentence. “Reaction” is not the best word here. Authors talk about the reflection of vegetation considering the visible and near infrared electromagnetic spectrum.

4.      L27. Do the authors mean “spectral resolution”?

5.      L 31. Revise the sentence. …especially the atmospheric one.

6.      L 35-41. Revise the sentence. I suggest to insert “or” before illumination geometry (L40).

7.      L 57. Do the authors mean “sometimes”?

8.      L 88-89. Do the authors mean “planting date”? I suggest “sowing date”.

9.      L110-112. Revise the sentence. I consider the word “but” (L111) is not necessary.

10.  L 132. Begin the sentence with capital letters (i- With).

11.  L 132. Do the authors mean “a sample of a plant”?

12.  L133-134. This sentence is not necessary. This information can be read in the equation 1.

13.  L 136. Revise the sentence. …, Nl the number of…

14.  L139. Homogenize the way the equations are referred in the text (eq. 2).

15.  L143. Explain the elements of the equation 3 in the same way than authors explained in equation 2.

16.  L 165. Do the authors mean “continue developing”?

17.  L180. Revise the style of the Figure 3 caption (size).

18.  L 222. Figure 4 should be Figure 5. There are two Figure 4 in the manuscript.

19.  L 233. Figure 5 should be Figure 6.

Next figure does not show any caption. Additionally, the title letters are in different style (font). 

Author Response

Dear Sir,

Thank you for your support and efforts. Your comments have improve my paper significantly. I followed all your remarks to correct all mistakes. Please note that all corrected mistakes and added remarks are written in red. Some redundant information and/or references are also removed.

Best regards

Round  2

Reviewer 1 Report

The authors implement a very interesting tool. The tool has a good potential to be used as an online learning tool. After revising it for the second time, the addition of some ground pictures (photos) would make the paper even more understandable and attractive. However, there is not a clear compromise to release the system in the web besides a short mention in Lines 205-206. A preliminary version could be available already. Is it not possible? Why? It would be a pity to have the paper published and the tool not published at all.

Some other concerns:

L107: spell out VNIR and SWIR;

L113: correct unit to mm^2;

L300: add a reference for CHRIS/PROBA (meaning and major paper describing it); add a mention that it reflects nadir viewing (if applicable);

L303: same above for Hyperion;

Author Response

Dear Sir/Madam,

Thank you one more time for your efforts and your helpful remarks.       

 Best regards

The authors implement a very interesting tool. The tool has a good potential to be used as an online learning tool. After revising it for the second time, the addition of some ground pictures (photos) would make the paper even more understandable and attractive. However, there is not a clear compromise to release the system in the web besides a short mention in Lines 205-206. A preliminary version could be available already. Is it not possible? Why? It would be a pity to have the paper published and the tool not published at all.

A-    Thank you for your kind and helpful remarks. A version 1 of the tool is provided with this paper with sample SQL database of the captured data such as multiple dates of spectral signatures for potato, onion
and wheat (Zip file). Hopefully, as indicated in the text (L231 to Line 233) the tool will be deployed for public use on the Web.

Some other concerns:

L107: spell out VNIR and SWIR;

A-    Words are spelled thank you

L113: correct unit to mm^2;

A-Corrected thank you

L300: add a reference for CHRIS/PROBA (meaning and major paper describing it); add a mention that it reflects nadir viewing (if applicable);

A-    Information
are added from Line 299 to 307 in red thank you for your helpful remarks.

Reviewer 2 Report

I have reviewed the resubmitted manuscript data-505923 (revision 2) and the authors have clarified all the questions and comments suggested in the previous revision. Thanks for the efforts to improve the manuscript.

I only have a minor comment detailed below:

       Minor comment:

Although authors have explained more specifically the data collection process, there are no information about the quantity of spectral signatures collected. In the manuscript, authors mentioned “many samples”, but in order to be sure that the amount of information is enough to consider it representative, more specific information about data collection should be shown (e.g. number of plots of each crop studied and number of measures collected).

Author Response

Dear Sir/Madam,

Thank you for your helpful remarks and comments. Please find my answers to your questions below:

I have reviewed the resubmitted manuscript data-505923 (revision 2) and the authors have clarified all the questions and comments suggested in the previous revision. Thanks for the efforts to improve the manuscript.

-Thank you for your help

I only have a minor comment detailed below:

       Minor comment:

Although authors have explained more specifically the data collection process, there are no information about the quantity of spectral signatures collected.

A-    Thank you for your remark. More information are added in Lines 177 to 179 in red color.

 In the manuscript, authors mentioned “many samples”, but in order to be sure that the amount of information is enough to consider it representative, more specific information about data collection should be shown (e.g. number of plots of each crop studied and number of measures collected).

A-    Thank you for your helpful remarks. More information are added in Lines 143 to 146 and Lines 164 to 65 in red color.
